# Comprehensive Examination of the Determinants of Damage to Houses in Two Wildfires in Eastern Australia in 2013

**Owen F. Price [1,*], Joshua Whittaker [1], Philip Gibbons [2] and Ross Bradstock [1]**

[1]  Centre for Environmental Risk Management of Wildfire, University of Wollongong, Wollongong, NSW 2522, Australia; Josh.Whittaker@rfs.nsw.gov.au (J.W.); rossb@uow.edu.au (R.B.)

[2]  Fenner School of Environment and Society, Australian National University, Canberra, ACT 2601, Australia; Philip.Gibbons@anu.edu.au

*  Correspondence: oprice@uow.edu.au; Tel.: +61-(0)-2-42-21-54-24

**Abstract:** Wildfires continue to destroy houses, but an understanding of the complex mix of risk factors remains elusive. These factors comprise six themes: preparedness actions (including defensible space), response actions (including defence), house construction, landscape fuels, topography and weather. The themes span a range of spatial scales (house to region) and responsible agents (householders through government to entirely natural forces). We conducted a statistical analysis that partitions the contribution of these six themes on wildfire impact to houses, using two fires that destroyed 200 houses in New South Wales (Australia) in October 2013 (the Linksview and Mt York fires). We analysed 85 potential predictor variables using Random Forest modelling. The best predictors of impact were whether the house was defended and distance to forest toward the direction of fire spread. However, predictors from all four of the other themes had some influence, including distance to the nearest other burnt house (indicating house-to-house transmission) and vegetation cover up to 40 m from the house. The worst-placed houses (undefended, without adequate defensible space, with burnt houses nearby and with a westerly aspect) were 10 times more likely to be impacted than the best-placed houses in our study. The results indicate that householders are the agents most able to mitigate risk in the conditions experienced in these fires through both preparation and active defence.

**Keywords:** bushfire; fire impact; fire weather; wildfire risk; property loss

## 1. Introduction

Wildfires continue to destroy buildings in fire-prone regions throughout the world. For example, in 2017, 7500 houses were destroyed in California [1], and 500 in Portugal [2], while Australia experienced its worst event in 2019/20 (3113 houses lost [3]). These are tragic events, but also opportunities to learn ways to reduce the level of destruction in future wildfires. To this end, there have been many studies of the determinants of wildfire damage, mostly in Australia [4–10] and the USA [4,11–14]. Both empirical post-fire research and physical models of fire impact have revealed a long list of potential determinants, which can be grouped into six main themes:

(1)  Preparedness actions. This is management of fuels up to 30 m around the house, commonly referred to as the creation of 'Defensible Space' [13];
(2)  Responsive actions. These are primarily actions taken to defend the house from attack;
(3)  House construction (materials, shape, gaps);
(4)  Landscape fuels (i.e., beyond the Defensible Space, >30 m from houses);
(5)  Topography;
(6)  Weather.

The six risk themes span a gradient of spatial scale, from individual houses through gardens, landscapes and regions (Figure 1). They also span a range of responsible agents. The householder is responsible for preparedness actions, though government agencies

may assist with advice, and in some cases, may enforce compliance. For example, New South Wales in Australia has a Hazard Complaint processes whereby residents can be required to remove fuel from their property. Responsive actions (defence) are sometimes carried out by firefighters, sometimes by householders and sometimes by both, though fire agencies offer no promise of protection to householders. House construction is partly a choice by householders, but there are government-mandated regulations covering aspects such as window material and gap dimensions. Fuel in the landscape is managed by people other than the householder (except for large private properties), and for communities in the Wildland Urban Interface (WUI), this is usually public agencies such as local or state government. Topography is a natural feature, but the extent to which houses are exposed to topographically derived wildfire hazard can be moderated by government regulation about where in the landscape buildings are allowed [15]. Weather is a natural feature that no human agents can control, although climate change is driven by human activity and in the long term, may increase the severity of fire weather [16].

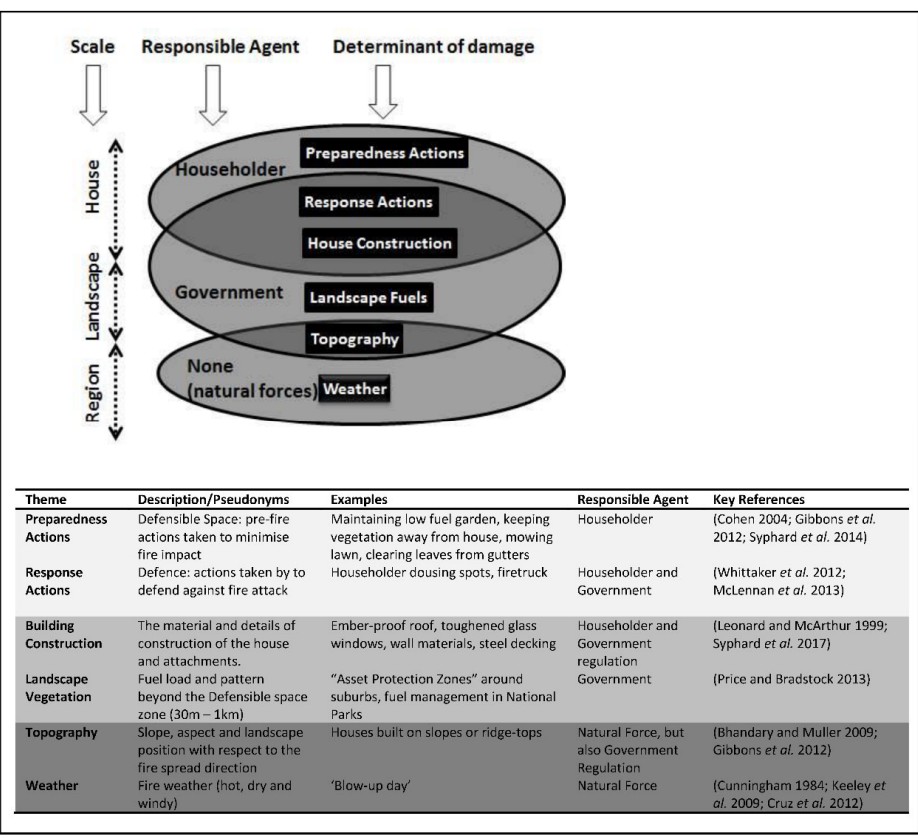

**Figure 1.** A conceptual model of the main determinants of house impact (called themes in the text), the agent responsible for mitigation and the scale at which they operate. The table provides a description, some examples of each determinant and key references.

The nature of fuels surrounding the house is critical to the survivability of a house exposed to a wildfire [17,18]. For example, vegetation overhanging the house, the height, density and distance to trees and other buildings, the presence of lawns and artificial fuels, such as fences, have been shown to influence house destruction [4,5,7,8,19–22]. The variety of factors involved is encapsulated by the concept of defensible space [4,12,17]. This is usually defined as the fuels in a zone up to 30–40 m around a house, managed so that the fuel load is low and there is sufficient horizontal and vertical separation between the fuel elements. Hypothetically, defensible space primarily acts to reduce flame contact and radiant heat from an approaching fire, but can also decrease the chance of spot fires igniting from embers, and it allows firefighters to operate

around the house in relative safety. Defensible space also equates to the radiant heat zone, the safe distance (approximately 40 m) for building survivability from radiant heat [13]. The concept of defensible space is enshrined in government regulations and guidelines for residents in many areas around the world (e.g., Anon [15] in Australia and California Public Resource Code Section 4291 in the USA).

Residents who defend their house can decrease the likelihood that the house will be destroyed [9,21–23]. Australian fire authorities support the right of residents to stay and defend their houses if they are sufficiently prepared [24]. Canada and the USA practice mandatory evacuation [25], such as the 2007 evacuation of 300,000 people in southern California [26] and similar events in 2018, while evacuation policy is less formal in most of Europe [27]. Even in Australia, the death of 173 people in the Black Saturday Fire of 2009 caused fire services to shift advice toward evacuating rather than defence, particularly in severe weather [28].

Several house construction factors have been found to be important, including wall and roof construction, whether the eaves are open, and the type, framing and amount of window [20,21,29]. Wooden roofs have high risk [22], but there are contradictory findings about the role of roof pitches and whether elevated houses are at greater risk [8,29]. Leonard and McArthur [30] concluded that the construction material was less important than the standard of construction. The importance of construction is reflected in legislative regulations covering house construction in fire-prone areas that define requirements for wall, roof and window design and materials depending on the expected level of heat exposure, and to minimise ember penetration into the house (e.g., Australian Standard AS3959 'Construction of buildings in wildfire prone areas'). However, these standards are not based on empirical evidence.

Fuels in the landscape (beyond 30 m from the house) can also have a major effect on outcomes, as estimated by the distance to native vegetation [31] to prescribed burning [5] or the extent of forest up to 1 km around houses [7]. The main rationale for landscape fuel treatment is to reduce the likelihood that the fire will spread to the Wildland Urban Interface, and it might be implemented as mechanical fuel breaks [32] or prescribed burning [33]. Landscape fuel treatments may also reduce ember attack at the edge of the interface, and embers are considered to be the dominant cause of house loss [30,34].

Topography and weather both affect the impact of fire on houses [5,22,35–37], and they are input parameters in all fire behaviour models [38,39]. Fire rate of spread increases when a fire burns up-slope [38], while slope and aspect affect fire severity (and inferred intensity) [36,40]. Fire weather indices such as the Australian Forest Fire Danger Index [41], the Canadian Fire Weather Index [42] and the US Fosberg Fire Weather Index [43] all contain components describing temperature, humidity and wind speed. Empirical evidence confirms the usefulness of these indices as predictors of several aspects of fire behaviour, including rate of spread [43,44] and severity [36,45].

Most of the historical research into wildfire impact has studied one of these themes in isolation, and a small proportion have studied two or more [5,7,14,20]. None have examined all of the themes at once. As a consequence, we do not know the relative importance of the factors (which ones to prioritise for mitigation programs), and we do not understand dependencies and interactions between factors (e.g., how does the effectiveness of preparedness vary according to weather?). For example, the contradictory evidence for different aspects of construction may be because the studies did not take into account the weather, which may alter the relative risk from different aspects depending on conditions.

In this study, we examined the impact on houses from two fires from the Blue Mountains (New South Wales, Australia) in October 2013, using Random Forest modelling to disentangle the independent effects of risk factors from all six themes on fire impact. This provided both a comparison of the importance of each theme and an assessment of the effect of each risk factor. By identifying the most important themes, we can reflect on who has most responsibility for mitigating risk to houses: householders, government or whether environmental conditions limit the ability to mitigate the risk. We also compared the Random Forest approach to the insights that would have been gained using univariate statistical methods (binomial regression), with the expectation that the Random Forest provides a more complete understanding of the various influences on fire impact.

## 2. Method

*The Study Area and Fires*

Our study focused on 540 houses exposed to two fires in the Blue Mountains of NSW in October 2013 (Figure 2), the largest property impact in NSW between the years 1994 [46] and 2019 [3]. The Blue Mountains are a typical Wildland Urban Interface, being both a large (19,000 km$^2$) area of eucalypt forest managed for conservation and the home for thousands of people. The 17th of October 2013 was a very hot and windy day. The Mt York and Linksview fires ignited 30 km apart from each other at 12:15 and 13:30, respectively, at the peak of fire weather severity (temperature ~32 °C, wind speed ~60 km/h and relative humidity 8%). The Linksview Fire was responsible for the largest loss of houses (195 destroyed plus 146 damaged). It ignited due to a branch striking residential powerlines within 50 m of property and within 5 km of 8000 houses (NSW Government cadastre, unpublished) situated among dry sclerophyll eucalypt forest. The fire burned 1768 ha on the first day and all of the damage to or destruction of property occurred then, although the fire continued to burn for four weeks. The Mt York Fire was similar, except for the fact that there were fewer houses exposed downwind from the ignition point and the elevation was greater (920 m vs. 320 m for Linksview). It burnt an area of 396 ha on the first day, destroying and damaging 10 and 3 houses, respectively, on that day, reaching a final area of 9400 ha. Although other bushfires occurred at the same time, we did not examine them because they caused fewer house losses (only 5% of losses between them) and the data were not consistent with the Linksview and Mt York fires.

None of the areas burnt by these fires had been treated with prescribed burning in the past 20 years, so the role of recent prescribed burning in mitigating damage or destruction could not be tested. The ignition point of the Mt York Fire had burned in a large wildfire in November 2006 (7 years prior), though the impacted areas had not. Most of the area impacted by the Linksview Fire had been burned by a large wildfire in December 2001, which destroyed eight houses. Houses since reconstructed on these sites were then re-exposed to the Linksview Fire, resulting in four being destroyed a second time.

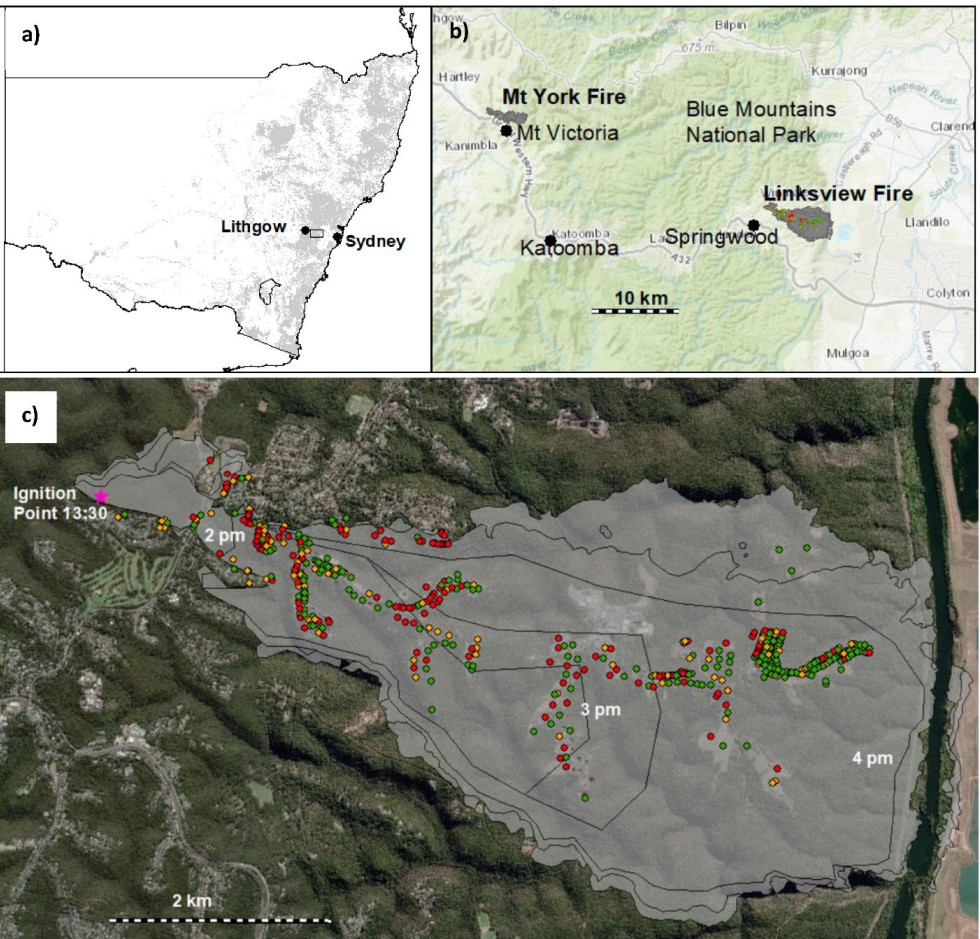

**Figure 2.** (**a**) The location of the study area (black outline) in NSW; (**b**) the study area showing the location of the two fires in the Blue Mountains region; and (**c**) the Linksview Fire showing the impact on individual houses. The background is an ESRI base map and the dark grey is the area burnt on 17 October 2013. Fire progressions are black lines with times in white text.

## 3. Data Sources

Data were obtained from six sources. In the days after the fire, a Building Impact Survey was conducted by trained staff of the NSW Rural Fire Service to record information about each exposed building (the Building Impact Survey). The survey was designed by CSIRO Land and Water, in consultation with the Rural Fire Service, and consisted of a detailed site visit lasting approximately 1 h at each building, with as many as 600 variables measured (depending on the nature of the house and damage). The surveyors strove to sample approximately one undamaged building for every damaged one, but did not include all potentially exposed houses. We filtered the survey to houses that stood within 100 m of the perimeter of the fires as they occurred at the end of 17 October when all of the damage had been inflicted, to remove houses that were probably not exposed. In practice, the survey was comprehensive in the centre of the fires, less so around the edges, and only 5% of the sample were outside that first-day perimeter. The survey examined many aspects of the construction of houses. This included the materials and configuration of the walls, roof, windows and doors; the nature of attachments such as decks and carports; the type, number and location of fuel elements in the Defensible Space zone (including cars, sheds, trees, shrubs, grass, garden mulch, gas bottles, door mats and timber); and evidence that houses were defended, whether by residents or firefighters. For example, there were nine types of cladding (exterior walls), but for 63% of houses, this was brick; lawn state could be 'green', 'semi-cured' or 'cured', and was 'green' for 40% of the 179 houses where it was recorded; there were five types of base

support (what the house rested on), and 61% of these were masonry. We selected 45 predictor variables using several criteria: the number of houses at which they were measured, their relevance to the study, removal of closely correlated variables and those with high potential for bias. Bias was apparent in some variables where destruction of the house obscured accurate measurement. For example, presence of a doormat was not used because houses with doormats showed lower impact than those without because doormats were often not apparent in destroyed houses. We also used GIS to generate additional spatial variables from the building survey data, including distance to the nearest house, damaged house, tree and fence. The survey was completed, comprehensively, for 540 houses. The variables we selected are listed and described in Table 1.

**Table 1.** The 84 attributes of houses used as predictor variables in the analysis, along with their source, associated theme, the number of houses with a value (n) and the percentage of deviance captured in a univariate binomial regression of house impact. Theme codes are PREP = preparedness action; RESP = response action; CON = construction; LF = landscape fuel; TOPO = topography; WEATH = weather.

| Variable (and Source) | Description | Theme | % Deviance | Sample Size |
|---|---|---|---|---|
| **Building Survey** | | | | |
| Base Supp | Most vulnerable support type? | CON | 4.00 | 181 |
| Cladding | What is the most vulnerable wall cladding material type? | CON | 3.63 | 540 |
| Deck Type | What was the decking board type? | CON | 4.36 | 110 |
| Deck Flamm | Does the deck including its support system contain any combustible elements? | CON | 2.40 | 134 |
| Deck Mater | What is the decking material? | CON | 0.05 | 95 |
| Sub Ht | Enclosure height above ground at highest point (cm)? | CON | 2.85 | 107 |
| Roof Mater | What is the roof material? | CON | 0.63 | 540 |
| Deck Roof | Does the deck have a roof? | CON | 0.10 | 540 |
| Base Type | Structure base type? | CON | 0.29 | 540 |
| House Fram | What was the house framing material? | CON | 0.45 | 424 |
| Roof Profile | What is the roof profile? | CON | 0.25 | 488 |
| Storeys | Number of functional levels? | CON | 0.37 | 540 |
| Timb Wind | Number of timber framed windows | CON | 0.26 | 295 |
| Gap Size | How big is the gap between the house base and enclosure? | CON | 0.12 | 85 |
| Carport | Is there a carport under the common roof? | CON | 0.07 | 540 |
| Base Encl | Is the subfloor enclosed on all side? | CON | 0.42 | 181 |
| Window Ht | What is the distance between the bottom of the window and ground? | CON | 0.06 | 289 |
| Garage | Is there a garage under the common roof? | CON | 0.01 | 540 |
| Lawn State | What was the state of the grass? | PREP | 4.05 | 179 |
| Carport Co | Were there combustibles in the carport? | PREP | 4.18 | 63 |
| Maint Lev | What was the overall maintenance level? | PREP | 3.59 | 540 |
| Grass Ht | What was the height of the grass (cm)? | PREP | 0.20 | 179 |
| Nrblddist | Distance to the nearest outbuilding (shed or garage) | PREP | 1.35 | 540 |
| Vegoverhan | Was there overhanging foliage? | PREP | 1.03 | 540 |
| Tree Bark | What type of bark does the tree have? | PREP | 0.76 | 124 |
| Gas Bottle | Is there a gas bottle? | PREP | 21.78 | 69 |
| Shed Combu | Were there other combustibles inside the structure? | PREP | 0.69 | 263 |
| Fence Mtrl | What is the object material? | PREP | 1.33 | 236 |
| Mulch Dpth | What was the depth of mulch (cm)? | PREP | 0.41 | 95 |
| Nrtre Dist | Distance to the nearest tree? | PREP | 0.46 | 540 |
| Fencedist | Distance to the nearest fence? | PREP | 0.70 | 540 |
| Nrshr Dist | Distance to the nearest shrub? | PREP | 0.23 | 540 |
| Veg Type | What type of vegetation? | PREP | 0.07 | 350 |
| Branch Ht | Height of lowest branch? | PREP | 0.00 | 167 |
| Tree Type | What type of tree is it? | PREP | 0.09 | 186 |
| Fence Ht | What is the height of the fence or wall (m)? | PREP | 0.01 | 540 |
| Vegtouchin | Was there foliage against the house? | PREP | 0.00 | 540 |
| Fence Open | What is the percentage of openings? | PREP | 0.08 | 130 |
| Defence | Was the house defended? | RESP | 6.22 | 540 |
| Burnhs_Dis | Distance to nearest burnt house? | RESP | 3.14 | 414 |
| Water Used | Was the tank or pool relied upon for structure defence? | RESP | 4.12 | 208 |

**Table 1.** *Cont.*

| Variable (and Source) | Description | Theme | % Deviance | Sample Size |
|---|---|---|---|---|
| Leave | When did the residents leave? | RESP | 8.82 | 44 |
| Water Type | What Type of Tank or pool was it? | RESP | 0.99 | 208 |
| Crew Supp | Did firefighters attend the house? | RESP | 1.68 | 507 |
| Person Day | Was somebody present on the day of the fire? | RESP | 0.40 | 55 |
| Crew Access | Is it easily accessibility for fire trucks? | RESP | 0.64 | 540 |
| Tankcapaci | What was the tank capacity? | RESP | 0.01 | 208 |
| **LiDAR** | | | | |
| Midcov_Over | LiDAR cover 4–8 m height overhanging the house | PREP | 1.56 | 414 |
| Upcov_Over | LiDAR cover > m height overhanging the house | PREP | 0.00 | 414 |
| Nearcov2 | LiDAR cover up to 50 cm height within 2 m of house | PREP | 1.90 | 414 |
| Elcov2 | LiDAR cover 0.5–4 m height within 2 m of house | PREP | 2.49 | 414 |
| Midcov2 | LiDAR cover 4-8 m height within 2 m of house | PREP | 2.18 | 414 |
| Upov2 | LiDAR cover >8 m height within 2 m of house | PREP | 0.27 | 414 |
| Nearcov10 | LiDAR cover up to 50 cm height 2–10 m of house | PREP | 2.90 | 414 |
| Elcov10 | LiDAR cover 0.5–4 m height 2–10 m of house | PREP | 2.07 | 414 |
| Midcov10 | LiDAR cover 4–8 m height 2–10 m of house | PREP | 5.46 | 414 |
| Upcov10 | LiDAR cover >8 m height 2-10 m of house | PREP | 1.76 | 414 |
| Elcov30 | LiDAR cover 0.5–4 m height 10–30 m of house | PREP | 0.80 | 414 |
| Nearcov30 | LiDAR cover up to 50 cm height 10–30 m of house | PREP | 2.40 | 414 |
| Midcov30 | LiDAR cover 4–8 m height 10–30 m of house | PREP | 3.04 | 414 |
| Upcov30 | LiDAR cover >15 m height 10–30 m of house | PREP | 0.61 | 414 |
| Midcov100 | LiDAR cover up to 50 cm height within 30–100 m of house | LF | 3.18 | 414 |
| Nearcov100 | LiDAR cover up to 50 cm height 30–100 m of house | LF | 3.62 | 414 |
| Elcov100 | LiDAR cover 0.5–4 m height 30–100 m of house | LF | 0.59 | 414 |
| Upcov100 | LiDAR cover 4–8 m height 30–100 m of house | LF | 0.83 | 414 |
| **Air Photos** | | | | |
| Distance To Veg N | Distance from the edge of each house to nearest woody vegetation in North compass quarter | PREP | 1.02 | 540 |
| Distance To Veg E | As above for East compass quarter | PREP | 1.41 | 540 |
| Distance To Veg S | As above for South compass quarter | PREP | 1.43 | 540 |
| Distance To Veg W | As above for West compass quarter | PREP | 1.02 | 540 |
| % Vegcover10 N | Visual estimate of % woody vegetation (tree and shrub) within North compass quarter with radius (10 m) from the centroid of each house | PREP | 1.74 | 540 |
| % Vegcover10 E | As above for East compass quarter | PREP | 2.92 | 540 |
| % Vegcover10 S | As above for South compass quarter | PREP | 1.16 | 540 |
| % Vegcover10 W | As above for West compass quarter | PREP | 3.69 | 540 |
| % Vegcover40 N | Visual estimate of % woody vegetation (tree and shrub) within North compass quarter with radius (10 m) from the centroid of each house | PREP | 1.48 | 540 |
| % Vegcover40 E | As above for East compass quarter | PREP | 0.35 | 540 |
| % Vegcover40 S | As above for South compass quarter | PREP | 4.19 | 540 |
| % Vegcover40 W | As above for West compass quarter | PREP | 5.27 | 540 |
| Cont For N | Distance from each house to nearest contiguous forest (from mapped source, Keith 2004) | PREP | 4.14 | 540 |
| Cont For E | As above for East compass quarter | PREP | 0.60 | 540 |
| Cont For S | As above for South compass quarter | PREP | 1.80 | 540 |
| Cont For W | As above for West compass quarter | PREP | 6.17 | 540 |
| **Topography** | | | | |
| Slope | Slope in degrees for a point 15 m west of house | TOPO | 2.17 | 540 |
| West-South-West | Was the aspect in the west-south-west quadrant | TOPO | 3.72 | 540 |
| **Weather** | | | | |
| Ffdi | Forest Fire Danger Index (Penrith Station half hourly) | WEATH | 0.00 | 539 |
| Windspeed | Wind speed (km/h) at Penrith (half hourly) | WEATH | 0.47 | 539 |

The role of active defence of the house was an important aspect of our study. We used multiple means to determine the binary variables whether defence of the house occurred, and whether defence by firefighters occurred. The Building Impact Survey reported physical evidence of defence, such as the presence of unrolled garden hose, sudden boundaries in the burnt area around the house and uncapped hydrants. Assessors often spoke with residents who were present at the time of survey, and key details of these conversations were recorded. The surveyors used all of this information to subjectively

assess whether the house was defended and whether by a firefighter or homeowner. The individual components of this assessment were not recorded. Most of the information on defence was from this survey, but it was cross-checked with other sources. In the month after the fire, formal interviews were conducted with residents at 40 of the houses surveyed in the Building Impact Survey and an online survey was returned by another five houses [47]. This process identified four additional cases of defence. In January 2017, we interviewed two firefighters who led the first teams that attended each of the fires. Using their testimony, aided by fire spread and impact maps and resource allocation numbers recorded in ICON (the official NSW RFS reporting system), we were able to reconstruct the general nature of the firefighting response.

Aerial photography with 50 cm resolution was obtained approximately one month before the fires by the NSW Department of Land and Property Information. Following the methods of Gibbons, van Bommel [5], this was used to visually estimate the distance to the nearest vegetation, contiguous block of native vegetation [48] (which in every case was eucalypt dry sclerophyll forest) and the percentage cover of vegetation within 40 m of each house for the four compass bearings. The variables are listed in Table 1 and the distribution of three of them are in Figure 3.

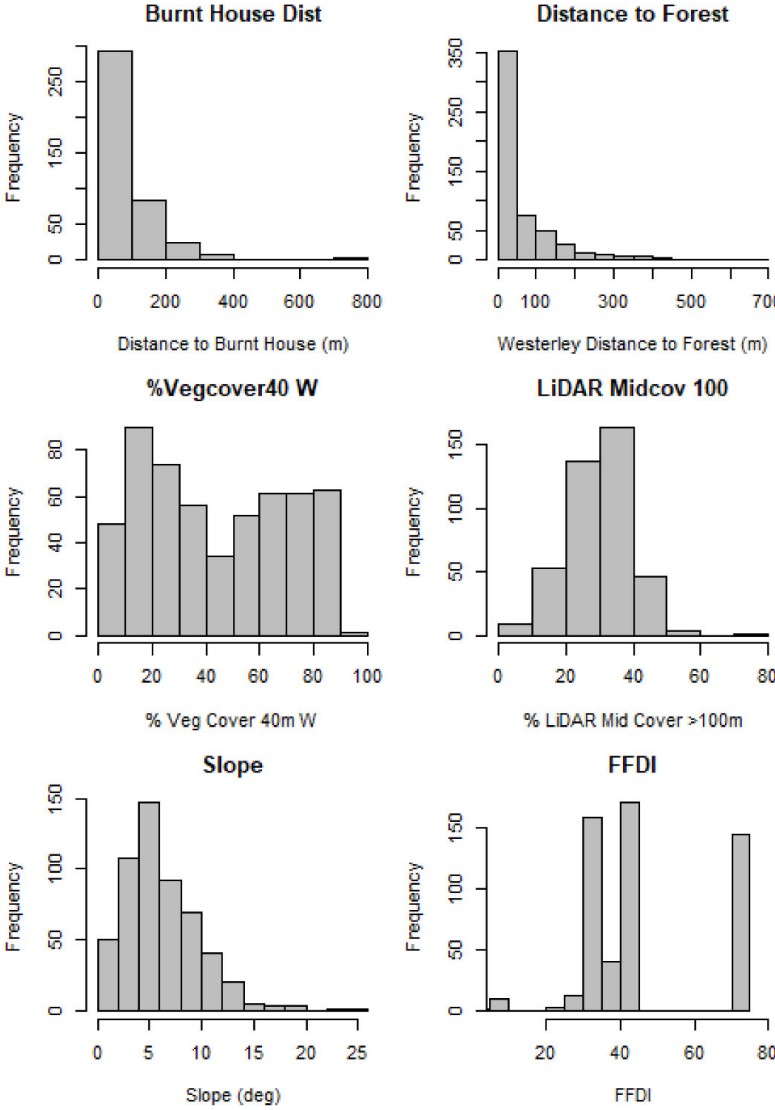

**Figure 3.** Distributions of values for six variables that feature in the final model or represent the themes. Refer to the text and Table 1 for descriptions.

Light Detection and Ranging (LiDAR) data were collected in the area around the Linksview Fire in February 2011 (20 months before the fire) by the NSW Department of Land and Property Information. The point data consisted of up to five returns per pulse and a mean point density of 1.05 m$^{-2}$. These data were processed using GIS to estimate the cover of commonly defined fuel strata (near surface, elevated and trees, [49]) in zones overhanging 2, 2–10, 10–30 and 30–100 m around each house, using the methods described in Price and Gordon [50]. Near-surface fuels are suspended above the ground to a height of 50 cm. Elevated fuels equate to the shrub layer and were defined as 0.5–4 m above the ground, and trees were divided into mid (4–8 m) and upper (>8 m) tree cover. LiDAR cannot, however, be used to estimate surface fuel (e.g., leaf litter) which is an important predictor of fire behaviour [41,49], so we had no surface fuel measures beyond the garden descriptions provided in other sources. There were 18 LiDAR variables (listed in Table 1, with one example distribution in Figure 3). The LiDAR distance zones corresponded to the defensible space theme, except for the 30–100 m zone, which was in the landscape fuel theme.

The NASA Version 3.0 Shuttle Radar Topography Mission 30 m digital elevation model (DEM) was used to calculate the slope and aspect of a point 15 m west of the house. This offset was used to describe the topography of the defensible space, rather than the house itself, which is usually on flat ground. Preliminary analysis suggested that aspect was best represented as the binary variable west-south-west (was the garden facing the compass quadrant from 202.5–292.5°, true or false?).

To estimate weather conditions at the time each house was exposed to the bushfire, we first created progression maps for the fires based on operational records from the Rural Fire Service and a search of social media platforms (Facebook and Twitter). This resolved six mapped time periods for the Linksview Fire and five for the Mt York Fire. Half-hourly weather data were obtained from Bureau of Meteorology weather stations (Penrith Lakes, 8 km from the Linksview Fire, and Mt Boyce, 7 km from the Mt York Fire). This was used to assign each progression period a value for wind speed and Forest Fire Danger Index (FFDI) [41]. At the time of ignition, FFDI was 75 and 29 at Mt Boyce (Penrith and Mt Boyce), which was in the 99.9th percentile for daily maximum FFDI at both locations. FFDI may have peaked at even higher values over the fire grounds due to wind gusts and the passage of a parcel of dry air around the time of the ignitions [51]. The weather eased in the afternoon and the minimum FFDI for exposed houses was 24 in the Linksview Fire and 7 in the Mt York Fire (Figure 2). According to Blanchi et al. [37], little house loss occurs in Australia with FFDI < 50.

## 4. Analysis

The post-fire building survey classified impact as 'untouched', 'superficial damage', 'light', 'medium', 'heavy' or 'destroyed. We defined the dependent variable as a binary variable ('impacted') with 0 for untouched, superficial or light damage, and 1 for medium, heavy damage or destroyed. Superficial and light damage usually involve only external features such as steps, decks or eaves and are easily repairable, so we considered them to be more closely aligned to untouched houses than destroyed ones. Since other studies place all damaged houses into the 'impacted' category [52], we tested this alternative classification, but found that it made very little difference to the results. Predictor variables were assigned to one of the six themes (see Table 1 for the list), and consisted of 85 variables: 19 construction, 49 preparedness actions, 9 response actions, 4 landscape fuels, 2 topography and 2 weather.

We used Random Forests [53] to identify the conditional effects of each theme (i.e., accounting for the other themes). Random Forest is an ensemble learning technique that builds hundreds of partial (or weak) regression tree snippets and combines them into a robust prediction model. Random Forest models are well suited to non-linear and interacting relationships and datasets with a large proportion of missing data such as ours [54].

We developed a Random Forest model from the full set of predictors using the party package in r software [55,56]. We used 2000 trees and assessed the model using variable importance (mean loss of accuracy), partial plots for the most important predictors and 'out-of-bag' accuracy testing. We used the model structure to define best- and worst-case situations and compared the actual level of impact for houses in these two cases. In order to compare the Random Forest method with simpler techniques, we also used univariate regression analysis to examine the relationship between damage and each of the predictor variables in turn.

The pattern of house loss has an element of spatial autocorrelation (similar outcomes are spatially clustered) for a variety of reasons. We addressed this first by controlling for weather (which shows strong spatial structure), and second by testing the degree of spatial autocorrelation in the raw impact score and in the residuals from the Random Forest model using Moran's I test.

Data and r code are available on request from the authors, though address and location information for the houses will be withheld for legal confidentially reasons.

## 5. Results

The Linksview Fire was responsible for 95% of the exposed houses, and analysis indicated a rate of spread of 3.1 km/h for the first 2.5 h, during which 91% of the houses were exposed. The Mt York Fire spread more slowly during that period, at a mean rate of 1 km/h. The fate of the 540 houses in the sample was 263 undamaged, 49 superficial, 44 light, 12 medium, 6 heavy and 166 destroyed. The range of the Forest Fire Danger Index (FFDI) experienced by 91% of houses in these fires was 34 to 75 (Figure 3). A total of 31% of the houses were defended. The median distance to continuous forest (toward the direction of the fire) was 25 m with a range from 0 to 668 m (Figure 3).

The Random Forest analysis identified defence as the most important variable affecting the likelihood a house would be impacted, followed by westerly distance to forest ('Cont Veg W'), distance to the nearest burnt house and west-south-west aspect (Figure 4). The proportion of defended houses that were impacted was 14.7% compared to 42.4% for undefended houses. Westerly distance to forest up to about 50 m distance from houses had a strong negative relationship on house impact. Forest cover beyond 50 m in this direction did not influence impact. Burnt houses closer than 40 m increased the likelihood of impact by approximately 10% (Figure 4). The model had 72.2% accuracy with 28% false negative and 28% false positive rates.

The impact on houses in the worst-case situation, as defined by the Random Forest model, was 71%. These were the 31 houses with forest <10 m to the west, undefended, facing west-south-west and with >50% vegetation cover within 40 m to the west. By contrast, only 3% of the 31 houses in the best-case situation were impacted. The attributes of best-case situation were the opposite of the worst case (>50 m from forest to the west, defended, not facing west-south-west and with <50% vegetation cover within 40 m to the west). Moran's I index of spatial autocorrelation was 0.113 for the impact score and 0.019 for the residuals of the Random Forest model, suggesting only a small amount of spatial autocorrelation.

We compared the Random Forest model to the insights that would have been gained by considering each predictor variable individually using binomial regression, similar to the approach with much previous house impact research [6,8,34]. More than half of the 84 predictor variables showed relationships with impact (capturing >1% of Deviance, Table 1). Eleven variables were shared among the top 20 ranks in both the Random Forest and individual models, including defence, vegetation cover 40 m to the west, the west-south-west aspect, wall cladding and distance to nearest burnt building (Table 1, Figure 5). The most common cladding (masonry) had an impact of 33%, while the highest impact was on mud-brick houses (impact 78%). The state of the lawn and the type of the base supporting the house were ranked highly in the individual models but not in the Random Forest, presumably because defence and vegetation cover account for their effect when all

factors are considered together. Presence of a gas bottle and deck type had large effects according to the individual models but were not in the Random Forest model, probably because the sample of houses was small for these variables (Table 1). Figure 5 shows the impact relationship for selected variables including distance to the nearest burnt house, westerly distance to forest, LiDAR mid-cover >100m, lawn state and wall cladding.

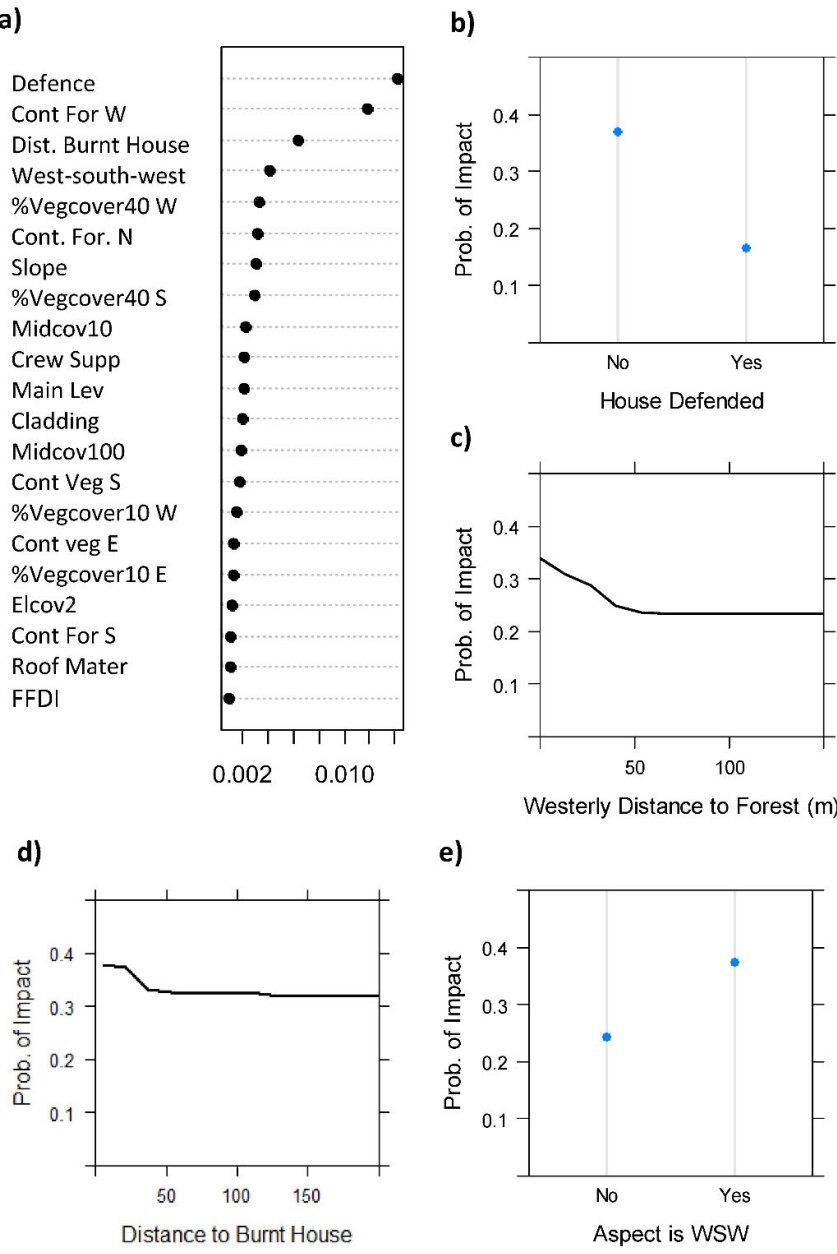

**Figure 4.** The Random Forest model for house impact comprising (**a**) the importance scores for the 21 best variables (scale is the loss of accuracy when each variable is dropped from the model) and (**b–e**) the partial effects plots for the four most important variables.

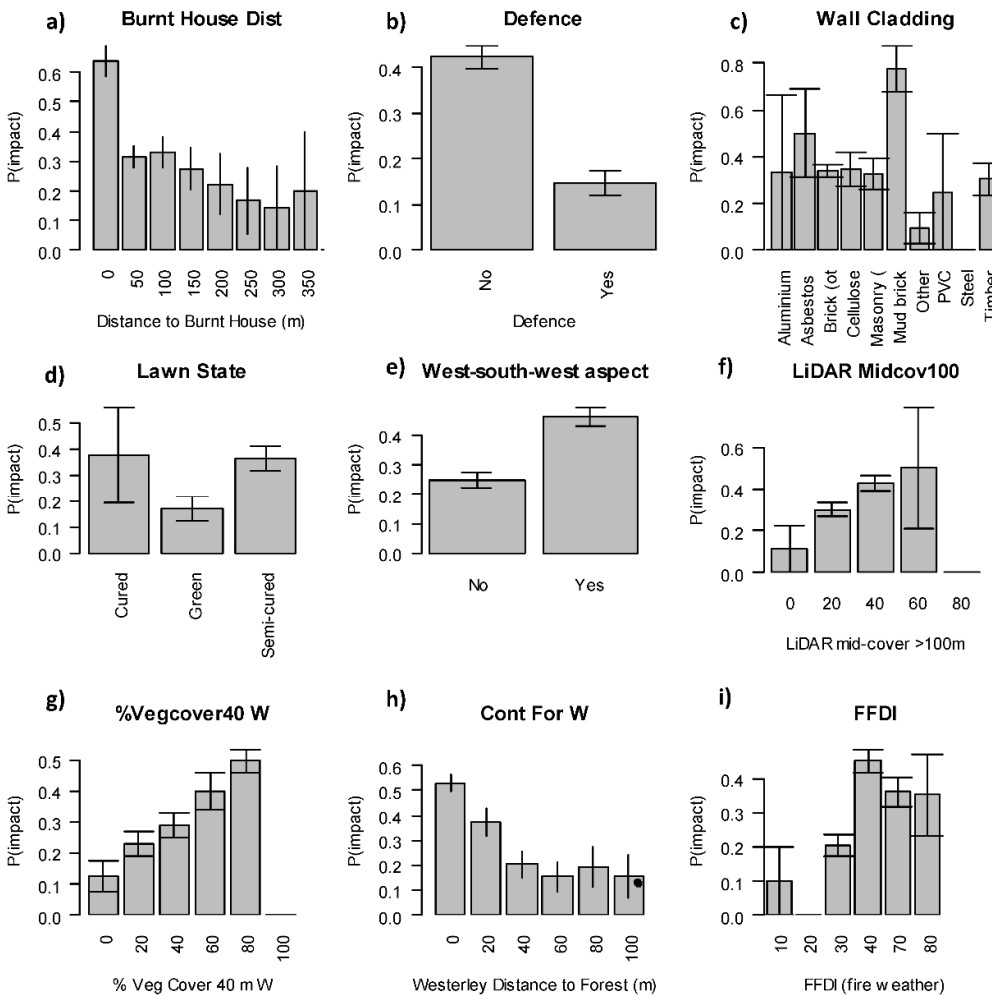

**Figure 5.** The proportion of houses impacted for selected predictors in the univariate analysis. Each plot (**a–i**) shows the proportion of houses impact for different levels of the selected predictor. Predictors were selected because they were important in the Random Forest or univariate models, or represented the six themes. The dot in (**h**) is the mean impact for cases with >100 m distance (*n* = 103).

## 6. Discussion

Our analysis has clearly identified preparedness, via the provision of defensible space (e.g., distance to forest), and defensive actions as the primary drivers of fire impact on houses. While many studies have shown such effects in isolation [6,8,12,21], our study provides a more complete overview of the relative importance of many potential determinants of house impact when exposed to wildland fires, compared with many previous studies. Provision of defensible space, as indicated by the importance of various vegetation metrics, and active defence once fires were close to property were clearly identified as having the most influence on impact.

## 7. Defensive Action

Houses that were not defended had almost three times the likelihood of being impacted, and house defence was selected as the most important variable in the forest model. Furthermore, distance to the nearest burnt house was identified as the third-largest risk factor, suggesting a substantial role of house-to-house transmission (also Figure 5a). This effect also highlights the importance of defence since it can help to stem the infection of fire observed among these houses. However, some caution is needed in this interpretation

of the influence of the nearest burnt house because it could also reflect strong spatial clustering in impact patterns due to other factors, such as weather. The benefit of defending houses has been demonstrated in several previous studies [9,21–23], including on the survival of nearby houses [52]. Our study reveals it to be the most important theme in these fires. The importance of defence raises questions about the policy settings for householder defence. In many countries, including the USA and Canada, complete evacuation is the official policy [26], and only in Australia is householder defence encouraged (in certain circumstances) [24]. However, even in Australia, there have been recent policy shifts toward a greater emphasis on evacuation [57]. The official ICON reporting system indicated that in the Linksview Fire, by 5 pm, when most of the damage had been inflicted, only 49 firefighters were battling the blaze, and the interviewed firefighter attested that only a small proportion of the exposed houses were attended (we estimated 16% of such houses). This relatively small response was because fire agencies were battling at least five other major fires, all of which ignited earlier. For example, the deployment at Linksview represented only 8% of the total deployment on the six fires, despite the fact that it suffered 89% of the property damage. This fire is an example of a fire where householder defence was vital for limiting losses. It follows that had the area been fully evacuated, the level of house impact could have been higher. However, this does not mean that householder defence should be promoted in all circumstances. For example, in the Victorian Black Saturday Fires of 2009, where 173 people died, defence was more risky, since 65% of those deaths were in the home [58]. Indeed, defence may become more dangerous as the fire weather becomes more severe [59]. The FFDI across the fires examined here ranged from 29 to 75, while FFDI values on Black Saturday were up to 189. Further research of the kind here is required to validate the general importance of defence by residents as a key determinant of house loss. Responsibility for the defence of houses is shared between residents and government agencies, and the balance tends to vary among countries [24,26].

## 8. Preparedness Actions

The second most important predictor in the Random Forest model was westerly distance to continuous forest (up to 50 m), reflecting the importance of defensible space. Several other defensible space variables ranked highly in the Random Forest model (including vegetation cover within 40 m, mid-level LiDAR cover within 10 m and maintenance level). Many measures of fuel can contribute to the quality of defensible space and several of these have previously been shown to be risk factors in one or more fires. These include the cover and type of plants in the garden [5,8,60], the presence of trees overhanging the roof [4] and lawns [6]. Several of these measures had modest effects in the univariate analysis, though were not ranked highly in the Random Forest model (including vegetation overhanging the roof, vegetation cover within 10 m of the house and the state of the lawn). Our results have major implications for policy and operational management in many fire-prone landscapes. They reinforce the benefit of current guidelines and regulations around the world that encourage residents to maintain defensible space. Maintenance of defensible space is primarily the responsibility of residents, so it follows that residents can substantially mitigate their wildfire risk under certain conditions. However, it should be recognised that the defensible space zone extends beyond the boundary of most residential properties, so to some extent, the responsibilities are shared with neighbours, including the managers of the wildland, in many cases.

## 9. House Construction

House construction variables were among the most significant in the univariate analysis, but none were selected among the top 10 in the Random Forest model. While the importance of house construction is recognised by the fact that governments impose legal construction standards on building in fire -prone areas, research into the specific features that contribute to risk has not always found consistent patterns [8,29]. In our study, wall cladding ranked 12th in the Random Forest model and 17th in the individual

models, but there were nine types of cladding with a range of impact rates that were hard to interpret (Figure 5c). It may be that the standard of construction (e.g., the size of gaps) rather than the actual materials that is the most important factor in determining vulnerability of structures [30]. Moreover, fire can damage houses through many pathways that may not be revealed through post hoc observations, particularly when houses have suffered major damage or are destroyed. This complexity is illustrated by the effect of gas bottles in our study. They were the highest ranked variable in the individual model, but presumably, because we only had data from 13% of houses, they did not influence the combined Random Forest model.

## 10. Landscape Fuels

The best landscape variable in univariate analyses was ranked 13th in the Random Forest model. Nevertheless, individually, both LiDAR derived cover of trees and shrubs at 100 m distance (Midcov100 and Elcov100) captured more than 2% of the deviance in impact in the univariate analysis (Figure 5f), and it is possible that the small number of landscape variables (four) used in the analysis has resulted in an under-estimation of the landscape effects. Certainly, other studies have highlighted an important role for vegetation management at distances of up to 1 km from houses [5,7].

## 11. Topography

Among topographic variables, west-south-west aspect was third ranked in the Random Forest model and slope was seventh. The influence of slope and aspect on various aspects of fire behaviour and impact have been highlighted on many occasions [36,61], but there have been other studies that found weak effects [5]. It is perhaps surprising that these variables were not more important. Possibly their effects were overwhelmed by the strong influence of preparedness and response actions.

## 12. Weather

Weather only had a minor role in determining which houses were impacted (Figure 5i), with FFDI being the 20th most important variable in the Random Forest model. Newnham et al. [62] also concluded that weather was not particularly influential in house loss in these fires. This is perhaps surprising, since studies have found that variation in weather during a fire influences several aspects of fire impact, including house loss [5] and fire severity [36,40,63]. While house impact was twice the rate at FFDI 75 than at 34 (38% impacted compared to 19%), it seems the weather effect was swamped by more important factors. Moreover, the muted role may be because this range of fire weather was quite low, for example, than that experienced in the Black Saturday Fires (FFDI range 5–189 [5]). In any case, weather probably played an important role in the rapid rate of spread, which itself contributed to the limited defence that was possible by firefighters. It is likely that in these fires, weather was responsible for exposing many houses to the fire due to its role in driving rate of spread, but other factors were more important in determining the fate of those exposed houses.

## 13. Other Factors

There were several variables that were not selected in the final model but were significant on their own and have been shown to be important in previous studies. Examples include distance to nearest building [5,7], vegetation touching the house and overhanging the roof [4], the presence of gas bottles [64], roof profile and angle [8,29] and decking material [10]. Several of these explained >2% of deviance in the univariate models, but were not selected in the multivariate models. Many of these are also mentioned by experts as risk factors [64]. The most likely reasons that many of these variables were not selected in the final model are: (1) they are of secondary importance to other factors (e.g., preparedness and response actions) and (2) they only come into play under certain circumstances because there are so many different ways that a fire can take hold. For example, the absence of trees

overhanging the roof in the final model does not mean that such trees are not risk factors, only that in some cases the tree did not come into play because the fire was halted by some other factor (especially defence).

The Random Forest was successful at identifying the conditional effects of the six themes. It was superior to a univariate approach because it controlled for the effects of other variables in the model. The univariate approach identified most of the 84 predictors as having some influence on impact, many of which are probably just the result of co-linearity among the predictors.

## 14. Limitations

There were a number of limitations and potential biases in the study.

- The study did not examine all potential risk factors and we were forced to discard many promising variables because of small sample size or potential bias. These included factors such as the presence of doormats or open windows and fuel loads further than 100 m from the house. This is why we focused on themes rather than the specific variables identified in the study.
- There is some uncertainty as to whether defended houses were attended by firefighters or residents because the criteria we used to discriminate relied largely on subjective 'signs' or the memories of interviewees who were present at only a small proportion of the houses. However, our estimates were corroborated by the Incident Reporting system that indicated a small contingent of firefighters were active in the Linksview Fire. It would be possible to be more definitive about what defence was carried out at each house and by whom by interviewing a larger number of householders and firefighters in the weeks after a fire (it is too late for this one). However, such a project would require a huge effort to obtain a sample as large as the one used here.
- Although our analysis suggests that structure-to-structure ignition was common, we do not know which houses ignited that way or whether the determinants of impact were different for houses exposed via burning vegetation or other burning houses. Further research on this phenomenon is needed because stemming this 'contagion' could dramatically decrease the impact of bushfires [65].
- There is some uncertainty about whether the houses in the sample are truly representative of the exposed houses. The RFS team focussed most of their effort on houses in the centre of the fires, rather than those on the edge. This probably reduces the potential bias, but those on the edge experienced variable and uncertain exposure. The statistical method and our inclusion of weather as a predictor controls for this variation in exposure to some extent.
- Being a statistical approach, this method overlooks risk factors that affected only a small proportion of the houses. For example, the presence of a stack of firewood next to a house is a known risk factor, but if only five houses had such a stack, our method would not identify it as important.
- Our analysis did not differentiate burnt and unburnt features as risk factors. The decision not to do so influences the results. For example, it is likely that knowing whether a tree overhanging the house was burnt would improve the accuracy of the model. However, our aim was to focus on risk factors that could be measured before the fire and so used in a predictive sense. The exception was distance to the nearest burnt house, which we did include because doing so identified an important risk factor.

## 15. Conclusions

This analysis of houses exposed to destructive fires in south-eastern Australia is the first to examine the relative importance of most of the known important influences on house impact. We found that preparedness actions and response actions were the most important themes (i.e., more important than house construction, landscape fuels, topography and weather). Preparedness is primarily the responsibility of the householder,

while response can be shared between householders and the fire agencies. Hence, the strong conclusion from this study is that actions by householders can strongly reduce wildfire risk to their properties.

While the distance to continuous vegetation was the most important of the preparedness variables, it is too simplistic to single this aspect out as the one needing attention, because many preparedness variables are correlated (for example, garden cover and vegetation overhanging the house). The most important message is that maintaining defensible space is important, and this involves reducing multiple sources of wildfire fuel around the house.

The strong influence of house defence should prompt discussion about the 'Prepare, stay and defend or leave early' policy that was widely applied in Australia [9,26]. It was clearly effective in these fires as it has been in others [9], and it may be useful in other countries. However, this is a complicated issue because active defence can increase the risk of death and its effectiveness may diminish with the severity of weather. It is therefore important to understand the circumstances where the potential benefit in property defence is outweighed by risk to life. The interactions among weather, human capacity and risks of defence are important considerations that need to be further understood [59].

While house construction, landscape fuels, topography and weather were not identified as important in these fires, they all had some influence and have been found to be important in previous fires. Hence, a focus on these elements in fire risk planning needs to be maintained. The principles derived from the models fitted in our study need testing more widely, especially in fires burning under more severe weather, different fuels and houses across a wider range of densities and with differing design features.

**Author Contributions:** O.F.P. conceived the study, collated data, conducted the analysis and led the writing. J.W. contributed advice about interview data, and assisted with writing and review. P.G. provided advice about measuring vegetation around houses and assisted with writing and review. R.B. assisted with writing and review. All authors have read and agreed to the published version of the manuscript.

**Funding:** This research was funded by the Rural Fire Service of New South Wales. P. Gibbons' participation was funded by Australian Research Council grant DP150100878.

**Institutional Review Board Statement:** This research was conducted under University of Wollongong Ethics Approval 2016/1017.

**Informed Consent Statement:** Informed consent was obtained from all subjects involved in the study.

**Data Availability Statement:** Data available on request due to privacy considerations.

**Acknowledgments:** We thank staff who worked on data gathering (Bronwyn Horsey, James Barker, Carrie Wilkinson, Josh Sharp-Heward, Ella Finney and Nina Price) and advisors from the RFS (Luke Catorall and Melissa O'Halloran).

**Conflicts of Interest:** The authors declare no conflict of interest.

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
