# Peer review of "Comprehensive Examination of the Determinants of Damage to Houses in Two Wildfires in Eastern Australia in 2013"

_fire, doi:10.3390/fire4030044_

Round 1

Reviewer 1 Report

I must say it was a pleasure to read this manuscript. The authors aimed to fill a gap in research by looking at all of the aspects that contribute to fire risk. Including weather, which is always mentioned when one doesn’t include weather in their analysis. I was pleased to observe that the conclusions the authors arrived at regarding weather related variables, match the arguments we’ve used before to explain why we didn’t include the, in the study. This was a necessary step to show once and for all that yes, weather is important, but highly difficult to measure at the house level and usually there isn’t enough variation among houses for the models to pick up differences. 

I liked how the authors used level of responsibility in their study. It is important to acknowledge that that are many actions homeowners can do and there are other that are shared. 

the text is clear, it has a nice flow and rather and I was very happy to see result from another country to align so well with results from other studies and with messages that so many fire researchers have been trying to bring awareness to for so many years. 

Congrats on a well designed and written study!

Author Response

Thank you very much for such a positive review.

Reviewer 2 Report

The paper gives relevant results about defensive and preparedness actions as predictor variables of houses damaged in wildfires. The use of a multivariable model is a contribution to this field of study.

The selection of six factors analyzed is grounded based on previous studies about the risk of wildfire damage.

The sources of information and techniques of statistical analysis are described in depth. The data analysis is precise. The authors transparently expose their interpretations and even possible biases in their views, opening a space for discussion. The limitations of the result are clearly described.

As a recommendation, a more depth description of house construction, landscape fuels, topography, and weather to the two cases analyzed can help understand why these factors are not relevant as predictors of damage.

Author Response

Thank you for your positive review. You have made one recommendation, which we wholeheartedly agree with. However, with so many variables involved, it is challenging to find the right level of detail. We have decided to make several minor changes, which we hope will convey enough information to address your recommendation:

  • In the methods we added more description for some of the variables, mostly for construction, but also preparedness and vegetation.
  • We illustrate these with a new figure (figure 3) showing the distribution of values for 6 selected variables.
  • We have added 3 more variables to the response plots for the individual model section (for Distance to nearest burnt house, Wall Cladding and Lidar Mid Cover 10m, now figure 5).
  • Added some more description in the results section of the effects of specific variables that were not selected.
  • Added some detail in the discussion about variables that did not rank in the final model.